# Dark Adaptation and Its Role in Age-Related Macular Degeneration

**DOI:** 10.3390/jcm11051358

**Published:** 2022-03-01

**Authors:** Archana K. Nigalye, Kristina Hess, Shrinivas J. Pundlik, Brett G. Jeffrey, Catherine A. Cukras, Deeba Husain

**Affiliations:** 1Retina Service, Massachusetts Eye and Ear, Department of Ophthalmology, Harvard Medical School, 243 Charles St., Boston, MA 02114, USA; archana_nigalye@meei.harvard.edu; 2National Eye Institute, National Institutes of Health, Bethesda, MD 20892, USA; kristina.hess@nih.gov (K.H.); brett.jeffrey@nih.gov (B.G.J.); 3Schepens Eye Research Institute of Mass Eye and Ear, Harvard Medical School Department of Ophthalmology, Boston, MA 02114, USA; shrinivas_pundlik@meei.harvard.edu

**Keywords:** dark adaptation (DA), phototransduction, cone-rod break (CRB), rod-intercept time (RIT), age-related macular degeneration (AMD), subretinal drusenoid deposits (SDD), longitudinal monitoring, spatial gradient

## Abstract

Dark adaptation (DA) refers to the slow recovery of visual sensitivity in darkness following exposure to intense or prolonged illumination, which bleaches a significant amount of the rhodopsin. This natural process also offers an opportunity to understand cellular function in the outer retina and evaluate for presence of disease. How our eyes adapt to darkness can be a key indicator of retinal health, which can be altered in the presence of certain diseases, such as age-related macular degeneration (AMD). A specific focus on clinical aspects of DA measurement and its significance to furthering our understanding of AMD has revealed essential findings underlying the pathobiology of the disease. The process of dark adaptation involves phototransduction taking place mainly between the photoreceptor outer segments and the retinal pigment epithelial (RPE) layer. DA occurs over a large range of luminance and is modulated by both cone and rod photoreceptors. In the photopic ranges, rods are saturated and cone cells adapt to the high luminance levels. However, under scotopic ranges, cones are unable to respond to the dim luminance and rods modulate the responses to lower levels of light as they can respond to even a single photon. Since the cone visual cycle is also based on the Muller cells, measuring the impairment in rod-based dark adaptation is thought to be particularly relevant to diseases such as AMD, which involves both photoreceptors and RPE. Dark adaptation parameters are metrics derived from curve-fitting dark adaptation sensitivities over time and can represent specific cellular function. Parameters such as the cone-rod break (CRB) and rod intercept time (RIT) are particularly sensitive to changes in the outer retina. There is some structural and functional continuum between normal aging and the AMD pathology. Many studies have shown an increase of the rod intercept time (RIT), i.e., delays in rod-mediated DA in AMD patients with increasing disease severity determined by increased drusen grade, pigment changes and the presence of subretinal drusenoid deposits (SDD) and association with certain morphological features in the peripheral retina. Specifications of spatial testing location, repeatability of the testing, ease and availability of the testing device in clinical settings, and test duration in elderly population are also important. We provide a detailed overview in light of all these factors.

## 1. Introduction

Dark adaptation (DA) refers to the capability to see in low light or darkness after exposure to bright light. This natural process also offers an opportunity to understand the functioning of cellular function in the outer retina and evaluate for presence of disease. This review summarizes dark adaptation measurement, along with key clinical findings in relation to AMD.

Excellent review articles about dark adaptation, written decades ago by Feldman and by Mandelbaum consolidated some of the early ideas about dark adaptation in humans along with the aspects involved in clinical measurement of DA [1,2]. Relatively recent papers delve into the physiology of dark adaptation with extensive research carried out focusing on the understanding of visual cycle, particularly functioning of photoreceptors and other retinal cells in the intervening decades. Notably, Lamb & Pugh [3] detailed the molecular mechanisms of the retinoid cycle and their relationship with observed visual function in humans and animals. Burns & Arshavsky [4] discussed the molecular and cellular bases of vertebrate phototransduction process and its role in photoreceptor functioning and retinal degeneration, while Tom Reuter [5] consolidated the last 50 years of progress of our understanding of the DA mechanisms. There are many excellent resources by eminent authors that describe visual changes during dark and light adaptation [6] visual function and functional vision in presence of aging and early AMD [7,8,9]. A recent literature review focuses on the effectiveness of current DA measuring methods in detecting onset and progression of AMD [10], however a specific focus on clinical aspects of DA measurement and its significance to furthering our understanding of AMD integrating physiological aspects of DA, AMD pathophysiology, significance of DA as a functional biomarker, different methodologies and correlation of DA with patient reported outcome measures (PROMs), has been an important area of development. Given the wealth of research studies in recent years specifically related to DA and AMD, this review article is timely.

First, we will briefly introduce dark adaptation and its physiology, discuss DA measurement and some of the practical aspects involved in DA measurement, and describe the various instruments used for DA measurement in clinic, along with some new technologies on the horizon, and certain potential strategies for evaluation. The second part of the review will cover histopathological and clinical findings regarding the role of DA in AMD.

## 2. Physiology and Measurement of Dark Adaptation

### 2.1. Dark Adaptation & Its Physiology

#### 2.1.1. Dark Adaptation

The ability to adapt to a vast range of changing illumination is one of the fundamental properties of human visual system, and this includes, to an extent, the ability to deal with both small and large increases or decreases in the light level. Our vision response depends on many factors including the magnitude and speed of this change, as well as the prevailing ambient illumination prior to the change. Here, we will focus on the transition from bright to low light condition or darkness.

One of the manifest observable responses to changing illumination is the change in pupil dilation. The diameter of pupil increases with decreasing illumination, and up to about a four-fold increase in the pupil diameter is possible from the undilated to fully dilated condition [11]. Initial pupil dilation in dark also happens relatively rapidly, typically of the order of a few seconds [12,13]. In addition to pupil dilation, another underlying response is that of our visual system: the sensitivity of our visual system increases over time in the dark (higher sensitivity means the ability to see lower luminance levels). However, compared to pupil dilation, the change in visual sensitivity happens much more gradually, typically of the order of many minutes [3,14].

Although pupil size is one of the important considerations when measuring dark adaptation [2,5,11] (its role is described in subsequent sections), we are more interested in the gradual increase in the visual sensitivity than the pupillary response when studying dark adaptation. Therefore, in the vision science literature in general, dark adaptation is defined as the slow recovery of the sensitivity of the visual system after exposure to very bright light followed by a rapid (sudden) transition to darkness, usually with a controlled pupil size [5].

#### 2.1.2. Characteristics of Cone and Rod Mediated Vision

Overall, human visual system is sensitive to luminance levels spanning about 10^14^ cd/m^2^ [6,15]. While this is a large range, the modulation is split between cone and rod photoreceptors. Cones function at the higher end of the luminance range (10 to 10^8^ cd/m^2^), typically referred to as photopic vision. It has been mentioned that there is no real upper operating limit for cones [14], but a luminance level of 10^8^ cd/m^2^ can be considered a safe upper limit (higher luminance values can be experienced if looking directly at sun, however it is accompanied with its own well-documented adverse effects on the photoreceptors). Cones are less sensitive, but faster in their response to the changes in the light levels [16]. In photopic conditions, rods are saturated. Between luminance levels of 10 to 10^−3^ cd/m^2^ both cones and rods are active, for what is known as mesopic vision. Towards the lower end of the luminance range (10^−3^ to 10^−6^ cd/m^2^), only rods are active, and this is referred to as scotopic vision. Therefore, rods are more sensitive than cones—with the ability to respond to even a single photon [17], but scotopic vision has lower spatial and temporal acuity [6,14,18], as well as lower chromatic discrimination [19].

#### 2.1.3. Phototransduction

Simply stated, phototransduction refers to the process of converting light to visual signals, which occurs in the retina. Given that many in-depth review articles dedicated to vertebrate phototransduction have been published recently [3,4,20], we will only provide an overview of this process to aid understanding of the dark adaptation response observed in humans and why it might be impaired in AMD. The following description is more applicable to the functioning of the rod photoreceptors and refers to the classic rod visual cycle.

Compared to typical neurons, functioning of photoreceptors is reversed: here a stimulus reduces the cell’s response or firing rate. In the dark, or resting phase, the photoreceptors are depolarized, which means the photoreceptor membrane potential is less negative (around −40 mv) [21]. A small percentage of cyclic GMP is bound to the sodium channels near the outer segment to keep them open, so extracellular sodium ions move in; and near the main cell body in the inner segment, the potassium channels are open through which the potassium ions move out, establishing a current circulating through the photoreceptor membrane to keep it depolarized.

Rhodopsin, the photopigment essential to phototransduction, is a tube-like seven-helix G-protein coupled receptor transmembrane protein (‘opsin’) with the light sensitive 11-cis retinal chromophore conjugated to a lysine residue of rhodopsin and oriented horizontally in the disc membrane of photoreceptor outer segments to optimize photon interactions. When light strikes a rhodopsin molecule, the bent 11-cis retinal isomerizes to linear all-trans-retinal, losing its color, in what is known as bleaching of the photopigment (photobleaching). This activates rhodopsin and sets off a cascade of reactions leading to a drop in the concentration of cyclic GMP causing cyclic GMP to unbind from the sodium channels, causing their closure and the hyperpolarization (membrane potential becomes more negative) [3] of the photoreceptor membrane—which defines the cell’s electrical response [22].

When hyperpolarized, the photoreceptors stop releasing glutamate. This signals the ‘ON’ bipolar cell and deactivates the ‘OFF’ bipolar cells. The bipolar cells then pass along the signals to ganglion cells. Rods synapse on rod bipolar cells that do not connect directly to the ganglion cells. Instead, they connect to the cone ON bipolar cells via amacrine cells. This means that whenever sensitized, the rods can take over the cone visual system at very low light levels—where the cones are not working [14,23].

Recovery of photopigment refers to the recycling of the all-trans retinal back to 11-cis retinal. This process has many steps (Figure 1), and they occur in the outer segments and outside of the photoreceptors, primarily in the RPE cells. All-trans retinal is not light sensitive and needs to be recycled back to make 11-cis retinal, which is light sensitive. This recycling is known as the retinoid cycle and it forms the basis of measurement of the recovery of the retinal sensitivities after bleaching in dark adaptation measurement. So, after photobleaching, the time required for replenishing light-sensitive 11-cis-retinal can be considered as the rod recovery time, which can be as about 40 to 50 min in rods after full photopigment bleach [3].

#### 2.1.4. Summarizing the Process to Explain the DA Response

The observed visual response related to dark adaptation can now be pieced together based on the characteristics of photopic, mesopic, and scotopic vision, and the phototransduction and retinoid recovery mechanisms. After exposure to bright light and transition to darkness, the cone sensitivity recovers faster, however it cannot go beyond a threshold value corresponding to the later part of mesopic vision, so the earlier part of the DA response is guided by the cone response, which plateaus. Rods that are saturated by the exposure to bright light, because of the photopigment bleaching, take longer to recover and start to become active only during the early mesopic stage. Slow rod recovery is because of the presence of free opsin following rhodopsin bleaching [24]. Over time, as their sensitivity recovers, the rods take over the cone pathway (piggybacking) and the DA response is essentially driven by the rod response (scotopic vision), until the rods reach their absolute threshold sensitivity. Since a Muller cell-based visual cycle is believed to be able to sustain cone function and account for rapid recovery of cone visual pigment after bleaching for continuous daylight vision, the measurement of rod-mediated adaptation (and associated curve–parameters) is thought to be more relevant to pathology associated with the RPE Bruch’s membrane complex and photoreceptor function [25,26].

### 2.2. Importance of DA Measurement

Earliest observations regarding the DA process have been documented in the late 18th century and early 19th century [27]. Since then, our understanding of the DA mechanism has increased tremendously. During the past few decades there has been an added focus on the clinical applications of the DA visual function.

There were early observations that photoreceptor sensitivity recovery was slower, along with general impairment of DA response in people with various diseases affecting photoreceptor outer segments, RPE, Bruch’s membrane (BrM) or choriocapillaris including AMD as detailed in subsequent sections.

Owsley et al. [28,29] have shown that rod function is particularly impaired in AMD. In addition, it was also shown that kinetics of the DA process (i.e., the rate of photoreceptor sensitivity recovery) was impaired, which means that delays in adaptation are associated with and sensitive in reflecting the presence of AMD. Jackson et al. [30] showed that kinetics of the DA process can be measured in relatively short time for clinical applications, such as screening for AMD, unlike measuring absolute threshold values of rod and cone sensitivities, having long test duration. Obtaining DA kinetics instead of absolute rod thresholds was shown to be a more sensitive, easier, and more feasible option in clinical setting as test could be completed in shorter duration, instead of having the patients sit in the dark room for 30 to 40 min, which is of significance given the elderly population affected by AMD [30,31,32]. However, the real significance of DA as a visual function test in the context of AMD was that the DA impairments were often seen in older individuals with normal macular health but with known AMD risk factors [33], before any significant decline in visual acuity, or sometimes even before any structural changes were noticed in the retinal imaging [34]. This aspect is discussed in detail below in Section 3.1.2 and Section 3.2.2 of this review.

### 2.3. Measurement of Dark Adaptation

#### 2.3.1. Two Approaches of DA Measurement

There are two main approaches of DA measurement: electrophysiology and psychophysical. Electroretinograms (ERGs) provide an objective measure of the activity of retinal cells in response to light flashes by recording the current generated across the retina via attached electrodes. As this measurement is done without requiring a behavioral response from the subject, it is particularly suited to studying functioning of photoreceptors or ganglion cells in animals. The early influential studies to chart the relationship between retinal sensitivity and rhodopsin recovery after bleach relied on the ERG response in rats [35,36]. ERGs are used in the clinic as well but almost exclusively to test final threshold responses after full adaptation which has particular use in approaching suspected cases of inherited retinal diseases [37].

In human subjects, using psychophysical approach to measurement of sensitivity is a viable alternative. This approach generally involves asking a subject to indicate their response (either verbally or by pushing a button) when presented with a visual stimulus. In this review, we will focus on psychophysical measurement of dark adaptation in humans.

#### 2.3.2. Overview of DA Measurement Procedure

Despite some variation in measurement protocols, the basic procedure involved in the psychophysical DA measurement process is similar. The goal is to measure the recovery of visual sensitivity in the dark after bright light exposure and therefore the measurement is carried out in a dedicated dark room. In practice, recovery of visual sensitivity means the ability to see stimuli of progressively lower luminance levels over time. The measurement apparatus primarily consists of a fixation target, a bright light source, and a testing light (stimulus). Typically, a single eye is tested at a time, with the fellow eye patched. To measure sensitivity recovery in dark, the first step, known as bleaching, is to expose the test eye (either the full retina or, for some AMD studies, a small region on the macula) to bright light. This step is intended to bleach photopigment. The bleaching step is followed by presenting a test stimulus in the location that received the bleach to record how long it takes for the photoreceptors in the bleached area to regain their sensitivity to low luminance stimuli.

The measurement starts with presenting stimulus of luminance value lower than the bleaching light level, but still many log units higher than the absolute threshold of the rods. The luminance level of the stimuli is changed according to a predefined staircase algorithm, depending on the subject response to the presented stimuli. The subject indicates if they see the presented stimulus (for example, by pressing a button) and the time (after bleaching) and stimulus luminance level (or the corresponding sensitivity level) are recorded. Thus, stimulus presentation continues contingent upon how the subjects respond to seeing the presented stimulus, until the specified end point criterion is met.

The DA response thus recorded consists of tuples of time and sensitivity (generally recorded over logarithmic scale). This DA response is also known as DA characteristic curve. Various parameters can be extracted from this response curve, which are informative about the DA status for the measured eye (Figure 2).

#### 2.3.3. DA Outcome Measures

The DA response or DA characteristics curve can be directly visualized and compared between different conditions. However, for more quantitative analysis, DA parameters, the metrics derived from curve-fitting dark adaptation sensitivities over time corresponding to specific cellular functioning are extracted from the DA curve. Some of the commonly used parameters that can be extracted from the DA characteristic curve are: absolute sensitivity thresholds for cone (plateau threshold) and rod phase (rod final sensitivity) of the curve, time to cone-rod break, cone recovery rate, rod recovery rate, among others (Figure 2) [28,29,39,40]. Parameters such as the cone-rod break (CRB) and rod intercept time (RIT) are particularly sensitive to changes in the outer retina [3,22,24,41].

Different kinds of curve fitting approaches have been employed to extract these parameters. A combination of linear segments S1, S2, S3 was used to fit to the sensitivity recovery curve for cone and rod components [24]. This scheme was refined in various studies of aging and AMD where DA measurement was performed in elderly subjects [29,39]. Further custom models were built to obtain the DA parameters by fitting a combination of linear and non-linear functions to different components of the DA curve [40,42,43].

Some of the recent DA instruments, such as AdaptDx (Maculogix, Harrisburg, PA, USA), produce DA measurement defined as the rod-intercept time (RIT), which is the time required for the rod sensitivity to recover by 3 log units. This outcome measure was determined to be effective in reliably and reproducibly quantifying the rod response, which is typically the focus of clinical DA measurement applications [31]. RIT as a parameter is very effective in understanding the kinetics of the DA process and a larger value of RIT indicates delays in DA corresponding to impairments due to the underlying conditions.

One of the limitations of using RIT as the outcome measure of DA response is that in some eyes with impaired DA, a valid RIT value is not obtained even after a long test duration, as the sensitivity never recovers to the RIT criterion. Thus, RIT cannot be used to track longitudinal changes in DA in certain cases, such as AMD patients. Therefore, using both additional curve parameters and additional curve metrics such as area under DA curve (AUDAC), are additional approaches to quantify the DA response. In addition to being correlated with RIT and offering similar diagnostic sensitivity, AUDAC can also reliably quantify the DA response in eyes where a valid RIT value was not obtained and can be used for follow-up measurement in these eyes with delayed dark adaptation [44,45].

#### 2.3.4. Measurement Conditions & How They Affect DA Response

Various measurement conditions affect the resulting DA response. Therefore, despite general similarity in the overall measurement procedure, details of the measurement protocol can affect the resulting DA curve. We will discuss some of the important measurement conditions and their potential effects on the resulting dark adaptation curve.

Bleaching: The amount of photopigment bleaching is one of the most important factors that can profoundly affect the shape of the resulting DA characteristics, with stronger bleaching resulting in longer recovery times as well as a more well-defined differentiation between the cone and rod components of the DA curve [3,46]. Stronger bleaching can be obtained by having a stronger bleaching light source, bleaching for a longer duration, or both. A very strong bleaching light can accomplish the same amount of photopigment bleaching in a fraction of a second as a long duration bleaching step with a lower luminance light source [46].

Luminance Range of Test Stimuli: Luminance range of the test stimuli can be important to ensure that rod response is measured. Since rods are saturated for light levels above the mesopic range and the absolute cone threshold is close to 10^−3^ cd/m^2^ [6], the ability of the apparatus to reliably measure rod response depends upon the presentation of test spots that are dimmer than 10^−3^ cd/m^2^. There might be situations where there are limitations on the lowest luminance levels that can be achieved, and therefore only a part of rod response can be obtained [47]. Alternatively, low luminance levels of the test stimuli can be achieved by using an appropriate combination of neutral density filters [40].

Wavelength of the Test Stimuli: Depending on the goals of DA measurement—whether the focus is to obtain rod response or both rod and cone response, the wavelength of the test stimuli needs to be chosen carefully. In people with normal color vision with three kinds of cone photoreceptors, the cone response can be obtained for wavelengths from the entire visible spectrum. Rods, on the other hand, are most sensitive to wavelengths near 500 nm (blue-green) [6]. Rods are not highly sensitive to wavelengths corresponding to red color [48], so typically fixation target is generally of red color and the commonly used test light is blue or blue-green, corresponding to the wavelengths where rod sensitivity peaks [49,50].

Location of the Test Spot: The foveola is rod-free [51], and therefore it is not possible to elicit rod responses for a centrally presented stimulus. Rods become more numerous just a few degrees outside of the fovea and in the retinal periphery leading to selection of test locations eccentric to the fovea. In AMD, the DA test location is chosen to be within the macula instead of far periphery to target the earliest locations of pathologic change [52]. Sensitivity recovery in rods is faster as test location is moved farther into macula in the eyes with significant AMD pathology [53,54], however there might be slight decline in sensitivity to detect AMD pathology as the test spot is moved farther into the periphery from 5° to 12° location. This is further elaborated in Section 3.3 of this article.

Spatial & Temporal Properties of the Test Spot: Spatial acuity is dependent on the retinal location as well as which sub-system, rod, or cone, is considered. Rods are absent in the foveola and are increasingly found in the periphery. Moreover, as we move toward the peripheral retina, the spatial acuity of rods is lower because of pooling [14]. This means that despite similar density, more rod photoreceptors pool their output to the ganglion cells. Therefore, to elicit a rod response, the test spot should be sufficiently large relative to the retinal location where the DA is being measured (suprathreshold). In various previous investigations, about 1.5° to 2° test spot is used [49]. As the stimulus size increases, the rod recovery becomes faster. Given that the rods also tend to have worse temporal acuity than cones [6], the test stimulus is typically shown as a flashing spot with a temporal frequency that is adjusted to elicit a rod response.

Other Details: A few other details about the measurement setup might affect the resulting DA response. Given that the rods have single photon sensitivity [5,14], the environment needs to be dark without stray light for reliable recovery of rod sensitivity. Stray light might lead to continued bleaching of rhodopsin and therefore the sensitivity might not recover to the expected threshold. Also, test termination criteria can determine the nature of DA response. DA measurement can continue either until the sensitivity recovers to a specific threshold value, or until a maximum time limit in dark, or whichever comes first [49].

#### 2.3.5. Physiological Factors Affecting DA Response

Pupil Size: Pupil size is the wild card when measuring DA, because pupil size directly controls the amount of light incident on the retina. It has been reported that pupil size can almost quadruple from an undilated state to a fully dilated state, which would make the pupil area 16 times larger, with a corresponding increase in the retinal illuminance and a change in the dark adaptation threshold of the order of 1 log unit [11,55]. Many of the previous lab studies about DA have tried to work around this variable by using apparatus that discount the effect of pupil size changes [5]. Some clinical studies involving DA measurement have relied upon artificially dilating the pupils before performing the measurement [34], although this does not completely account for the variation in the pupil size between subjects. However, the study population skews towards elderly in some of these studies that involve artificially dilating the pupil when measuring DA, and it has been shown that the elderly show lower variation in the change in pupil size per log unit luminance change [56].

Age: There is ample evidence that increasing age is related to delays and impairment in dark adaptation [7,39,57]. Specifically, elevated rod thresholds as well as delayed DA kinetics, have been observed [39]. The main reason for this observed impairment of DA with increasing age has been described as the net effect of various changes in the retina [7], reduced pupil size—both light and dark-adapted pupil size [56], and increasing opacities in ocular media. Because of the significant association of age with impaired DA, age becomes a confounding factor when studying DA in AMD patients, who tend to be above 50 years of age. A detailed discussion on aging and DA follows in Section 3.1 of this review article.

Presence of other ocular or systemic conditions: DA is impaired in many inherited retinal diseases (IRD) that cause the degeneration of rod photoreceptors, such as retinitis pigmentosa, and those diseases that primarily affect the Bruch’s membrane such as Sorsby fundus dystrophy, late-onset retinal degeneration (LORD) and pseudoxanthoma elasticum (PXE), among others [14,58,59]. Impaired DA was also shown in Stargardt disease; however the extent of impairment may not be as much as many of the other IRDs [3]. Other conditions such as vitamin A deficiency have also been shown to affect the DA negatively [3]. Hypoxia, particularly seen in diabetic eyes, could result in impaired DA [60].

#### 2.3.6. DA Measurement Instruments

DA measurement has been carried out in lab for research purposes using custom apparatus or modified perimeters [50,61,62]. For many decades, the Goldman-Weekers adaptometer was the only commercial instrument for DA measurement in humans. The Goldmann-Weekers adaptometer, which relied on mechanical movement of a drum and paper recordings has been superseded by adaptometers incorporating computer control and LEDs or computer monitors for stimulus presentation. Recent works also employ custom setup involving computer screens, bleaching flash light sources, and filters [40,63]. However, dedicated clinical instruments have been developed in recent years [64]. The DARK-adaptometer (Roland Consult, Germany), the MonCvONE (Metrovision, France) and Dark Adapted Chromatic Perimeter (Medmont, Australia) are examples of perimetry instruments that could be used to measure scotopic thresholds and dark adaptation. The AdaptDx (Maculogix, Harrisburg, PA, USA) is an instrument dedicated to measuring dark adaptation with an output of RIT. Table 1 lists instruments available for DA measurement along with some of their associated features. For example, some of these instruments require a separate bleaching apparatus and some only quantify part of the dark adaptation curve up to a criterion threshold.

In recent years, newer technologies involving mobile devices, virtual reality goggles, and AI have been introduced for measuring DA. With the goal of simplifying DA measurements to the possibility of self-testing, a mobile app for DA measurement was developed [47]. Preliminary evaluation showed that reliable DA measurement with modern smartphones is possible and further evaluation is ongoing. Also, virtual reality based goggles systems for DA testing including AdaptDx Pro (Maculogix, Harrisburg, PA, USA) and re:Vive 2.0 (Heru; Miami, FL, USA) have been recently released. These are summarized in Table 1.

## 3. Retinal Pathology and Dark Adaptation

As described in the sections above, dark adaptation in part depends on the biochemical recycling of all-trans-retinol back to 11-cis-retinal, which occurs across the interface of photoreceptor outer segments and retinal pigment epithelium (RPE). The supply of nutrients, including vitamin A and oxygen necessary for dark adaptation, must traverse through Bruch’s membrane (BrM) from the choriocapillaris. Thus, pathologic changes to any of these cell layers or inter-cellular spaces have the potential to impact dark adaptation [3]. Monogenic diseases affecting these structures including the anatomy of BrM (Sorsby Fundus Dystrophy [SFD], Pseudoxanthoma elasticum [PXE]), the function of enzymes in the RPE (RPE65-related Retinitis pigmentosa, Late-onset retinal degeneration [LORD]), or rod structure (*RHO*-associated Retinitis pigmentosa), all cause delays in dark adaptation [26,58,59,65,66,67,68]. Lowered concentrations of metabolites in the choriocapillaris (e.g., prolonged vitamin A deficiency) may also delay dark adaption. Age-related macular degeneration (AMD) is a complex disease that has pathologic changes in multiple cellular layers including the RPE and BrM, and also involves extracellular deposits on both sides of the RPE. Thus, AMD is a disease whose cellular pathophysiology may be probed with careful study of dark adaptation [69,70].

### 3.1. Histopathologic Findings in the Aging Retina and Associated Functional Alterations

#### 3.1.1. Histopathology

As AMD is a disease where age is the salient risk factor, understanding the effects of aging—independently from AMD—on dark adaption is a first step in using this measure to probe AMD pathophysiology. With increasing age, changes in ocular morphology include thinning of the retina, accumulation of lipofuscin within the RPE, thickening of BrM, accumulation of basal deposits on and within BrM, and accumulation of drusen [71], all changes predicted to affect dark adaptation. There is also histopathologic evidence of a specific rod vulnerability and subsequent greater rod photoreceptor loss compared to cone photoreceptors both in aging and AMD [52,72].

While cone photoreceptor count and distribution remain rather stable across the retina during the life span, the number of rod photoreceptors declines by about 30% from the second to ninth decade of life [72,73]. First rod loss is observable in the inferior retina; later, as a ring of the paracentral macula [72]. Aging effects to BrM are believed to underly the changes observed in rod photoreceptor density [72]. Since rod photoreceptors recycle their substrates mainly via the RPE, they are dependent on the molecular diffusion via BrM for rhodopsin regeneration [74]. Cone photoreceptors, on the contrary, are additionally supplied by Müller cells and therefore, cones are believed to be less affected by BrM alterations [26].

#### 3.1.2. Function

The aging changes observed in histopathologic studies in vivo result in delays of dark adaptation in older individuals compared to younger adults. Overall, there is a rightward shift of the DA curve with increasing age. The time to the cone-rod break increases about 39 s per decade. The slope of the second component of dark adaptation also decreases with age (0.02 logUnit/minute per decade) [3,75]. When comparing a group of participants in their 70s to participants in their 20s, the older group showed a delay to the cone-rod-break of 2.5 min and an overall delayed recovery to pre-threshold sensitivity values of about 10 min [39]. Several recent studies employing different test modalities measured dark adaptation from multiple retinal loci across the retina in aging eyes [76,77].

In addition to dark adaptation, another important psychophysical measure of rod function is scotopic retinal sensitivity. Pre-bleach scotopic sensitivity (or scotopic thresholds) is a measure of the minimum light intensity that can be detected once the retina is fully dark adapted. Decreased scotopic sensitivity (increase in scotopic threshold) implicates changes in rod function independent of the molecular mechanisms involved in dark adaptation. Like dark adaptation, scotopic sensitivity worsens with age (0.08 log/decade) and this aging effect is constant across the central retina [57,77,78].

### 3.2. Effects of AMD

#### 3.2.1. Histopathology

Histopathologic investigations of AMD donor eyes demonstrate overt pathologic changes with the deposits that accompany and define the clinical evidence of this disease process. Both, focal sub-RPE deposits (soft drusen) as well as more ubiquitous basal laminar deposit in BrM, are observed in even intermediate stages of AMD [79,80]. Additionally, deposits on the apical aspect of the RPE, defined as subretinal drusenoid deposits (SDD), occupy space in the interphotoreceptor matrix; A space that is normally free from material that could hamper the transport of cis-retinal from the RPE to the photoreceptor outer segment and the return of trans-retinol from the outer segment back to the RPE [3,80].

RPE cells comprise a vital role for outer retinal nutrient supply, retinal maintenance, and metabolism. In AMD, metabolic pathway dysregulation involving lipid synthesis and storage and impairments of the photoreceptor outer segment autophagy processes are evidenced as well as cell dysmorphia and alterations of intracellular granules compared to healthy RPE cells [81,82,83,84,85].

While the initiating events of AMD are still being elucidated, the most significant observable cellular change in post-mortem eyes is rod photoreceptor loss in the parafoveal area [86]. In AMD, both rod and cone photoreceptors are lost with a 30% to 40% reduction in photoreceptor density relative to age-matched controls [52]. Photoreceptor loss is most pronounced in a 0.5–3 mm parafoveal region resulting in the greatest loss in an annulus of decreased photoreceptor count in the corresponding 1.5–10° around the fovea. Within this parafoveal region, rod death precedes that of cones and rod loss occurs to a greater extent than for the cone photoreceptors [52]. Thus, the loss of rod photoreceptors in the paramacular region in AMD occurs over the same region of rod loss observed in non-pathological aging. Given the changes observed in aged eyes without AMD and the histopathologic changes observed in AMD, a model piecing together a possible sequence of events was proposed: aging alone, without RPE disease, leads to a certain amount of rod loss (with minimal cone loss), but in the presence of AMD, RPE dysfunction contributes to additional rod death and less so to cone death [52]. Given the pathologic consequences of AMD to the cells relevant to dark adaptation, it follows that the mechanisms underlying dark adaptation would be negatively affected by this disease.

#### 3.2.2. Functional Deficits in AMD

Early visual changes noted by AMD patients include difficulty with adapting to changes in lighting, particularly to function in low light environments, often noted when driving at night. These changes in visual function often are noted before morphological fundus changes are evident in AMD and well before visual acuity deteriorates [87,88,89]. Difficulty with seeing in low light is consistent with the histologic findings show early and preferential loss of rods over cones [86]. Psychophysical testing further supports selective rod vulnerability in AMD with impairment in both scotopic retinal sensitivity and dark adaptation in these patients [28,54,90,91]. Additionally, scotopic sensitivity is more greatly affected than photopic sensitivity in AMD [40]. Significantly, rod-mediated DA may be impaired even in early stages of AMD when patients still have normal visual acuity [29,34,42].

Owsley et al. 2001 [29] reported at least one DA parameter was impaired in 85% of AMD patients compared with controls. By comparison, only 25% of pre-bleach sensitivity measurements in the AMD group were abnormal. These results were consistent with an earlier study by Steinmetz [91] (1993) who found slowed DA in all 10 patients with age-related BrM changes while only four of these patients also had decreased retinal sensitivity. Subsequent larger studies have confirmed a greater effect on dark adaptation than scotopic sensitivity in AMD [54,92,93,94] Flynn et al. (2018) [54] used the disparate effects of AMD on scotopic sensitivity and dark adaptation to categorize AMD into three functional phenotypes. Subjects with (1) normal scotopic thresholds and normal DA, (2) normal scotopic thresholds and abnormal DA and (3) abnormal scotopic thresholds and abnormal DA [76]. These functional phenotypes provide an additional means of characterizing AMD separate to traditional structural classification (e.g., AREDS).

The full DA recovery curve is dependent on at least two underlying mechanisms, recovery kinetics and final thresholds for both the rods and cones. The time to the cone-rod break is dependent on the interplay between these rod and cone parameters. Tahir et al. (2018) [63] reported that the earliest DA abnormality observed in AMD was a reduction in the steepness of rod-recovery slope and delays in the time to the cone-rod break was not observed until more advanced disease. The delay in the CRB may be indirect evidence of elevation in cone-thresholds in advanced disease. The complexity of DA changes observed in AMD patients was illustrated by Dimitrov et al. (2012) [90] who highlighted patients with normal and prolonged cone-rod break and both, steep and shallow slope of rod-recovery.

#### 3.2.3. Association of DA Dysfunction with AMD Severity including Reticular Pseudodrusen

As DA has the potential to reveal changes relevant to the pathophysiology of AMD, it would follow that the eyes with more severe AMD changes would have more abnormalities on DA testing. Clinical evidence of AMD includes the findings of sub-RPE drusen, RPE pigment abnormalities, and reticular pseudodrusen (aka subretinal drusenoid deposits) and in late stages-geographic atrophy and secondary neovascularization [95]. Different staging systems have been introduced to classify AMD disease stage and its progression [96] (e.g., by the AREDS Research Group 1999).

Regardless of the AMD classification system used, increasing AMD severity is associated with increasing levels of impairment in rod-mediated scotopic thresholds and DA [42,54,90,97,98,99,100]. Cone- mediated impairment may increase with AMD severity [90] but this is not a universal finding [42].

Flamendorf et al. 2015 [98] evidenced a monotonic increase of the rod intercept time (RIT, i.e., delays in dark adaptation) in AMD patients with increasing disease severity (Figure 3). AMD patients with large drusen in both eyes and the patients with large drusen in the study eye and advanced AMD in the fellow eye showed significantly longer RITs compared to age-similar subjects without any fundus changes. The results held whether a modified simplified severity grade scale was used, or classically defined AMD severity scaling [101] supporting a true association robust to the exact severity scaling.

Dimitrov et al. 2012 [90] studied retinal function in 357 subjects using an expanded AMD grading system of 13 severity groups (taking into account the size of drusen, its distinctness and retinal hyper- and hypopigmentation as well as characteristics of the fellow eye). All functional parameters including those that describe DA increased consistently with increasing drusen grade and patients with pigmentary changes had worse DA than those without. However, the testing protocol used in this study exhibited a floor effect, and the most severe disease stages did not demonstrate further delays in rod recovery rates.

When discussing AMD characteristics affecting DA, reticular pseudodrusen (RPD) take a unique and extraordinary role, since they are independently associated with impaired dark adaptation [54,93,98,99,102,103]. Histologically referred to as subretinal drusenoid deposits (SDD), RPD have been identified as extracellular material anterior to the RPE [104]. Contrary to sub-RPE soft drusen, RPD deposits include membranous debris from photoreceptor outer segments, unesterified cholesterol, and complement [105,106]. Clinically, RPD comprises a specific phenotype that is associated with fast progression to late AMD stages [107,108].

Compared to eyes with soft drusen, eyes with RPD showed far more pronounced influence on DA impairment [54,98,99,102,103]. In studies including several AMD stages, the majority of RPD patients do not dark adapt or show such slow dark adaptation that they do not reach a pre-defined criterion dark-adapted threshold (RIT) within the 20–40 minute allotted testing times [54,93,98,99,102]. With extended testing times, Luu et al. (2018) [103] were able to show that retinal sensitivity within the central 4–6 degrees continue to recover for several hours in some RPD patients. Five of the six RPD patients did reach their pre-defined criterion dark-adapted threshold after 24 h of dark adaptation. In the one patient with the most severe RPD grade, central locations did not recover to criterion [103].

RPDs are not only a characteristic in AMD, but also in monogenic diseases with a primarily impaired BrM, such as PXE, SFD, and LORD [109,110]. These patients exhibit a pronounced rod-mediated DA impairment, too, suggesting a common pathologic pathway of functional impairment associated with the formation of deposits on the apical side of the RPE in the presence of a diseased BrM [59,67,111].

Additionally, there is evidence that morphological alterations even if located peripheral to conventional stimulus locations and peripheral to central (30°) fundus imaging, that are often used in retinal standard care and on which most classic AMD gradings rely, show an impact on dark adaptation. Peripheral alterations found in this context include reticular pigmentary changes and a mottled FAF aspect in the far periphery, associated with impaired dark adaptation. This suggests that pathologic effects outside the stimulus location and outside conventional AMD grading areas influence retinal function [112].

These findings reveal a correlation of impaired DA to AMD stages but limitations in assigning single or localized characteristics other than RPD.

#### 3.2.4. Functional Phenotypes and Spatial Effect across the Macula

While scotopic sensitivity is reduced and dark adaptation slowed with aging, these changes do not vary with eccentricity across the central retina [54,63,78,93]. In contrast, there is a marked spatial gradient in rod-mediated loss across the macula in AMD which becomes more pronounced with increasing AMD severity [28,54,63,91,93,102]. Scotopic sensitivity losses are most pronounced in the parafovea (4–6 degrees) at all levels of AMD. More advanced AMD and RPD may result in total loss of rods centrally and reduced rod sensitivity out to 12–18 degrees eccentricity [54]. Psychophysically, there is a spatial gradient in rod-dysfunction in AMD that mirrors the spatial pattern of rod loss observed in histopathologic exams of eyes from AMD patients.

Dark adaptation is affected along a steep gradient in AMD with slower dark adaptation occurring in the 4° and 6° [54,93,102]. Flynn et al. (2018) [54] defined a new parameter, RITslope to characterize the spatial change in RIT with eccentricity. In control subjects, RITslope was close to zero as RIT remained relatively constant from 4° and 12°. In AMD patients, RITslope increased with disease severity and was highest in RPD patients [54,93]. The experiments by Luu et al. (2018) [103] further highlight the spatial gradient in dark adaptation in RPD. While retinal sensitivity at most non-central loci recovered to a criterion threshold in 30 minutes in six RPD patients, dark adaptation at the central loci (4° and 6°) continued to recover for hours. Recovery to criterion required up to 24 hours of dark adaption for some RPD patients at the most central test points. Chen et al. (2019) [99] followed dark adaptation over 4 years in patients with AMD and RPD. In a subset of (mostly RPD) patients who showed no dark adaptation at the 5° test loci at baseline, they were able to measure and quantify change in dark adaptation at 12° test loci following a reduced bleach. Flynn et al. [54] also found that a considerably higher number of advanced AMD/RPD patients exhibited no dark adaptation from loci in the superior retina compared with the inferior retina. As a result, their DA analysis focused on measurements from the inferior retina (Figure 4).

These combined results highlight the importance of either measuring DA at multiple loci across the central retina and/or identifying the optimal eccentricity in patients with advanced AMD and RPD in particular. In addition to loci, the retinal hemisphere may be an important consideration in testing DA in RPD patients.

In this consideration, profound knowledge of the most sensitive testing location is needed.

#### 3.2.5. Structure-Function Correlation Using Multimodal Imaging

In addition to findings evident on clinical fundoscopy or fundus photography, an expanded range of AMD characteristics can be found on multimodal retinal imaging. With emerging new imaging modalities and instruments and higher resolution, further investigations of AMD characteristics and their correlation with functional output are possible. Hyperreflective foci (HRF) on optical coherence tomography have been shown to correlate with hyperpigmentation on fundoscopy and have been demonstrated to be an important risk factor in the 5-year progression of AMD [113,114,115]. They are believed to represent anteriorly migrating RPE cells and thus be associated with tissue activity that subsequently leads to disease progression [113]. Recently, smaller hyperreflective specks have been introduced as a more prevalent finding indicative of less severe pathology and present both in normal subjects and AMD [100]. Both HRF and HRS have been associated with delays in rod-mediated DA compared to AMD patients without HRF and HRS, adding to evidence that these findings represent risk factors relevant to AMD progression.

Optical coherence tomography (OCT) can also be used to quantify both the neural retina as well as the drusen deposits associated with the disease. The metric defined as the RPE drusen complex (RPEDC) has been defined as the distance from the inner aspect of the RPE down to the outer aspect of BrM, thus including any sub-RPE drusen material, and due to segmentation, also including subretinal drusenoid deposits (SDD) on the apical side of the RPE. Analyses of retinal function correlated with RPEDC demonstrated significant correlations between cone function and RPEDC and also longer DA testing times with greater abnormal thinning volume of the RPEDC. Further elucidating which pathologic change—RPE drusen collapse, RPE thinning, SDD presence—will help unravel the pathologic progression of the disease.

Supporting the RPE and photoreceptors, but not quantified in the RPEDC metrics is the choroid, which supplies nutrients, vitamin A, and oxygen to the outer retina. In a study on a young healthy cohort, no significant correlation of RIT and SFCT was identified [116]. However, subfoveal choroidal thickness (SFCT) has been reported to be negatively correlated with RIT in AMD with thinner choroidal measures demonstrating longer RIT [98]. The relationship between SFCT and DA testing requires multivariate analyses as choroidal thinning is also associated with the presence of RPD. In multivariate modeling of RIT SFCT shows no significant independent contribution to RIT after RPD presence is accounted for [98].

Studies investigating the local effects of pathologic changes under the test stimulus have indicated that the presence of pathology in that test spot space is associated with DA testing abnormalities, but that abnormalities in the macula, outside of the test spot, influence the dysfunction identified [98,112]. Further studies are needed to parse out the local and eye-based effects. It is possible that some of the dysfunction is conferred by the role of deposits at the level of the BrM that are not quantifiable and cellular changes that have yet to be measured.

#### 3.2.6. Dark Adaptation as an Early Marker in Eyes at Risk of AMD–Transition from Health to Disease

Some recent studies have suggested that DA testing is able to reveal changes that parallel the earliest stages of AMD (AREDS II severity scale two to four) and can discriminate them from sole aging effects [34,117]. Other functional testing such as best-corrected visual acuity, low luminance acuity, contrast sensitivity, and mesopic sensitivity have failed to be able to detect early disease stages [34].

Interestingly, eyes in normal macular health but with risk factors for AMD demonstrate abnormalities in DA assessment.

As previously outlined, BrM alterations are accepted to contribute to the AMD pathophysiology but are not readily visualized on fundoscopy or on state-of-the-art imaging. The utility of DA testing can be explored in early disease stages or even to detect early disease stages in AMD. In a study investigating participants with normal macular health, rod-mediated dark adaptation is impaired in 22% of subjects. When further investigating these patients, the DA impairment was associated to known risk-factors of AMD. Specifically, patients with impaired DA had a three- to four-fold probability of exhibiting elevated CRP, an independent risk factor for AMD development [33].

A study of genetic risk factors, as measured using a genetic risk score for AMD, and threshold sensitivities in patients without overt retina structural abnormalities, was performed, finding no significant correlation between genetic test score and scotopic threshold sensitivities [118]. However, it has yet to be determined whether genetic risk score may confer a risk of DA abnormalities. More work needs to be done to elucidate the potential relationship between AMD genetic risk factors and DA testing.

Congruously, longitudinal investigations revealed that healthy eyes with a delayed RIT at baseline were at higher risk for developing AMD after three years than age-similar eyes with normal RITs [34].

These findings indicate a very fluent gradient between aging, subclinical disease, and manifest disease. Furthermore, DA is a measurement able to monitor the transition from a healthy but aged retinal status to (pre-clinical and clinical) disease stages, which will be of high value in identifying patients at risk and in early disease stages for upcoming therapeutic trials and potential therapeutic interventions in the future.

#### 3.2.7. Longitudinal Comparison of Morphological and Psychophysical Disease Progression

Comparing psychophysical measurements longitudinally and determining meaningful change of values demands profound knowledge regarding the repeatability of the technique and specific instrument used. RIT as one of the currently established outcome measures shows repeatability coefficients between 4.4 min (AdaptRx device) and 7.6 min (Medmont dark-adapted Chromatic Perimeter), considering that these studies also differed regarding their sample sizes and testing locations [38,98]. Another study found a coefficient of variation of 15% for the rod recovery slope and 10% for the final threshold after 30 min [119]. Other parameters, such as cone threshold (3.9 dB) final threshold (5.3 dB) and rod recovery (0.54 min/degree) showed comparable reproducibility. However, a considerable better repeatability coefficient in the inferior hemisphere (RC = 3.5 dB) compared to the corresponding superior testing point (RC = 6.6 dB) warrants further investigation [38].

Since psychophysical alterations in aging and AMD form a continuous spectrum and constitute a sequence of events in diseased subjects, DA testing is also an attractive measure for monitoring longitudinal disease changes.

Some studies that suggest theDA metrics can change significantly over the timeframe of 12 months even without observing parallel changes in fundus appearance have demonstrated measurable change in DA psychophysical measures [31]. A 2-year follow up study identified a mean RIT prolongation of 10.5 minutes and an increasing percentage of patients exhibiting a minimal prolongation of 6, 3, and 1 minutes, respectively. This finding exceeds by far the cross-sectionally determined prolongations in DA due to aging [39,77].

Furthermore, a 4-year follow up study revealed associations of RIT prolongation to disease severity, both at baseline and after 4 years. Of note, patients developing RPD exhibit a significantly increased RIT prolongation after 4 years than patients without RPD [99].

A 2-year longitudinal study of eyes with AMD demonstrated that the DA function slows in most eyes over the 2 years and that smoking was a significant risk factor, not only to the risk of AMD, but to progression of delays in DA. All studies are united in their conclusion of that DA has the potential to be a suitable measure for monitoring the disease progression in interventional trials, since DA impairments occur parallel to or even earlier than progression on conventional to severity grading on fundoscopy or fundus photography.

#### 3.2.8. Role of Genetic and Environmental Factors in Dark Adaptation

As a multifactorial disease, AMD is dependent on both environmental and genetic factors. From a comparison of monozygotic and dizygotic twins, a strong genetic influence on the cone-mediated functions but less on the rod-mediated functions could be evidenced [120].

The strongest genetic associations in AMD in general can be attributed to polymorphisms in the CFH and ARMS2 gene, contributing to an increased risk of developing AMD. However, healthy subjects without any funduscopic characteristics of AMD carrying these pathogenic polymorphisms exhibited prolonged RITs. This effect is further pronounced when investigating AMD patients. For both groups, subjects without fundus changes and AMD, the effect was larger for ARMS2 than for CFH polymorphisms. Additionally, the effect was stronger for homozygous than for heterozygous affected subjects [121].

Since the evaluation of phenotypes was one of the limitations of this study, it should be taken into account that the ARMS2 polymorphism resulted in a specific phenotype (e.g., with pronounced deposits at the BrM) and subsequently in the aforementioned functional deficits. This is further supported by a high correlation of the individual genetic risk score with disease stage after correction for age [118].

Despite accurate AMD classification systems and DA paralleling disease stage and progression, there are interindividual differences in DA among AMD patients of the same disease stage. Congruously, environmental factors modulating DA function in AMD could be determined after correction for age and disease stage [122]. First, a family history of AMD was a risk factor for longer RIT, indicating on the importance of genetic contributions as discussed above. Secondly, a higher body mass index is contributing to decreased adaptation, hypothesized to lead to a chronic inflammatory status that demands more vitamin A and reduces vitamin A availabilities in some organs [123,124,125]. Thirdly, higher (self-reported) alcohol intake was found to contribute to delayed DA. The pathophysiologic explanation is that ethanol is suspected to be a competitive inhibitor of retinol in retinal tissues [126,127].

Contrarily, cigarette smoking, which is accepted as an additional risk factor for developing AMD, has been found not to be a risk factor for delays in DA [33,128,129].

Since environmental factors—opposite to genetic factors-—can be targeted by lifestyle changes and therefore potentially modulate the disease, further longitudinal and interventional studies on these and other risk factors are needed.

#### 3.2.9. Relevance to Daily Life–Patient-Reported Outcome Measures

Establishing DA as an objective psychophysical test that reflects AMD disease pathophysiology is an enormous step in the development of outcome measures for earlier stages of AMD. However, it is also important to consider and investigate how these measures, and in particular, dysfunction, impact the participant’s activities of daily living. Despite a good visual acuity, these impairments can be not only uncomfortable, but in some situations, dangerous (e.g., tunnels when driving). Hence, subjective evaluation of visual function under low light conditions are important when evaluating the disease stage as well as for potential treatments of these symptoms (e.g., Vitamin A substitution) [130,131]. Furthermore, the FDA has expressed a need for patient-reported outcomes in clinical and interventional trials. Early studies of DA incorporated self-reported questionnaires to capture difficulties with day-to-day activities as measured by the Activities of Daily Vision Scale (ADVS) score [132]. Different questionnaires have since been explored to reflect daily situations under low light conditions, of which the 32-item Low Luminance Questionnaire and the 10-item Night Vision Questionnaire are the most prominent candidates [131,133,134].

A tremendous influence on the patient’s life quality in AMD and other diseases could be attributed to DA [89,133,135]. Specifically, rod-mediated DA, represented by a prolonged RIT, and other low-luminance function parameters correlate to the patients’ subjective visual function and quality of life. All subcategories of the LLQ were significantly correlated to RIT, however the strongest was “driving”, which implies an important role of functioning DA in these subjects given the high relevance of mobility in aging [89].

### 3.3. A Note Regarding the Choice of Instrument and DA Testing Protocol Utilized

As we have presented, when targeted to areas of the retina with earliest pathologic involvement, DA testing has the potential to reveal very mild dysfunction, but that pathologic changes associated with intermediate AMD can confer such severe dysfunction that certain eyes reach the DA test upper limits, making discrimination of the more severe stages difficult [76,93,103]. Different testing strategies can be optimized to maximize the utility and informative value of DA testing, with the understanding that all choices come with trade-offs. While single spot testing is easiest to administer, it forces a choice of spatial test location. Locations close to the fovea may be best suited for highest sensitivity to detect abnormalities, whereas more eccentric locations would result in DA tests where the desired sensitivity is reached in a shorter time, and few eyes would reach the upper limits of the test [54,93] (Figure 4). Testing with instruments and paradigms that could test several test points at once would enable the capture of several spatial locations in a given test and allow for calculations of spatial slopes of DA metrics (e.g., RITslope), which is a metric that could better define the DA kinetics of a study eye [76]. These instruments currently are more limited to niche research groups and are more difficult to administer and to interpret results from. In addition, by testing recovery of dark adaptation to several spots, the number of sensitivity points over time for any one test location are more sparser, making curve-fitting more challenging.

The choice of bleach can also be modulated to adjust the DA curve parameters, e.g., using a reduced bleach when investigating RPD groups, allowing for any recovery within a reasonable testing time.

Forethought as to the intent of the study, the population of interest, and the duration of the study are all relevant to the design and choice of the particular DA test used in a clinical study.

### 3.4. Use of Dark Adaptation as a Screening Tool for AMD

Two studies have investigated the clinical use of the AdaptDx for diagnosis and prognosis of AMD. In a study utilizing a short (≤6.5 min) DA test in participants with intermediate AMD and no AMD, a binary interpretation of the DA test was found to have a 90% diagnostic sensitivity and specificity for diagnosing AMD compared to gold standard color fundus grading [31]. A subsequent longitudinal study suggested that aged eyes with abnormal DA are more likely to develop AMD (as graded using color photographs) 3 years later [34]. Whether functional DA changes precede or predict structural AMD changes is under current investigation using additional imaging modalities such as OCT [26,58,59,65,66,67,68].

Several DA instruments now have FDA approval or clearance for use in clinical DA testing. DA manufacturers have entered the commercial market and promote their instruments to be used outside of clinical trials. (https://www.maculogix.com/amd/dark-adaptation-testing/, last accessed on 10 January 2022) (https://www.seeheru.com/technology/, last accessed on 10 January 2022) Recent developments include the size and interface of the devices, e.g., as augmented reality glasses (AdaptDx Pro, Maculogix, Harrisburg, PA, USA) and multiple additional functions (Heru re:Vive 2.0, Heru INC., Miami, FL, USA). There is also a Current procedural terminology (CPT) code for dark adaptation testing (CPT92284) which is an active code given by the center for Medicare & Medicaid services (CMS) Reimbursement, making reimbursement for DA testing possible under certain indications in the United States including AMD, night blindness, and hereditary retinal dystrophy. The reimbursement for AMD is payor specific by insurance providers; however, most payors will pay one occurrence per visit without the use of laterality modifiers.

## 4. Summary Paragraph

Since AMD is the leading cause of blindness in industrial countries with no effective therapy to date, initiating meaningful interventional trials remains one of the primary challenges among retinal scientists [136]. However, even with promising approaches emerging, the need for AMD-specific endpoints reflecting early and intermediate disease stages in interventional trials becomes even greater. DA reflects dynamic changes in rhodopsin regeneration. For rods, regeneration is dependent on functioning visual cycle enzymes, molecular diffusion via BrM and sufficient nutrient supply by the choroid [3]. Hence, DA is a measure that specifically mirrors ultrastructural changes of the pathophysiologic key tissues in AMD in very early and partially subclinical disease stages. The instrumentation to test DA has seen tremendous development from initial time-consuming experimental instruments and protocol to instruments for clinical use. Different protocols available for screening for AMD and tracking disease progression can help use DA as an important parameter in clinical trials. The testing of rod-based DA parameters in the form of RIT is available with evidence on the reliability and test-retest repeatability for AdaptDx (MacuLogix, Harrisburg, PA, USA), using focal retinal bleach of 76% at a single predetermined superior retinal location 5 or 12 degrees from the fovea [31,53,98], versus Medmont Dark-Adapted Chromatic Perimeter (Medmont, Nunawading, Australia), using a pairing of an full field 30% bleach external to the instrument, with eight testing locations from fovea straddling the vertical meridian [38]. The measurement of rod-based DA kinetics as a functional parameter is of particular relevance given the pathology of AMD primarily affecting rod outer segments, RPE, and BrM, with the involvement of choriocapillaris, as it provides structure function correlation. DA therefore is a suitable outcome measure for interventional trials in AMD [137].

## Figures and Tables

**Figure 1 jcm-11-01358-f001:**
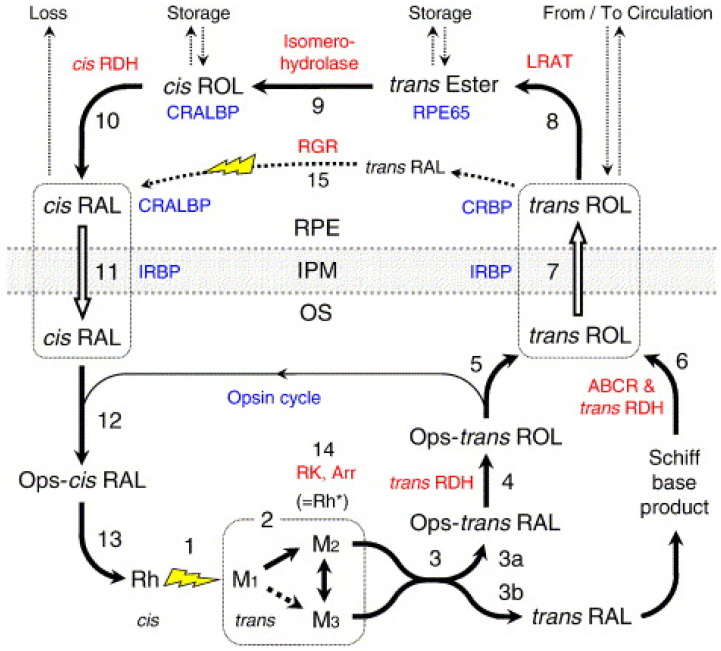
The retinoid cycle. Reproduced with permission from Lamb et al. 2004 [3]. represents important steps, enzymes and chaperon proteins comprising the retinoid cycle. (1) Photoactivation; The 11-cis retinal isomerized to the all-trans-retinal [(=Rh*) represents activated rhodopsin after absorption of photon] (4) reduction of aldehyde, (7) transport of all trans retinol across photoreceptor plasma membrane and the interphotoreceptor matrix (IPM) to RPE cell chaperoned by Inter-photoreceptor retinol binding protein (IRBP); (9) Isomerization from the all-trans to the 11-cis form, (10) oxidation of 11-cis retinol to 11-cis retinal and its delivery to opsin in the photoreceptor outer segments across the IPM.

**Figure 2 jcm-11-01358-f002:**
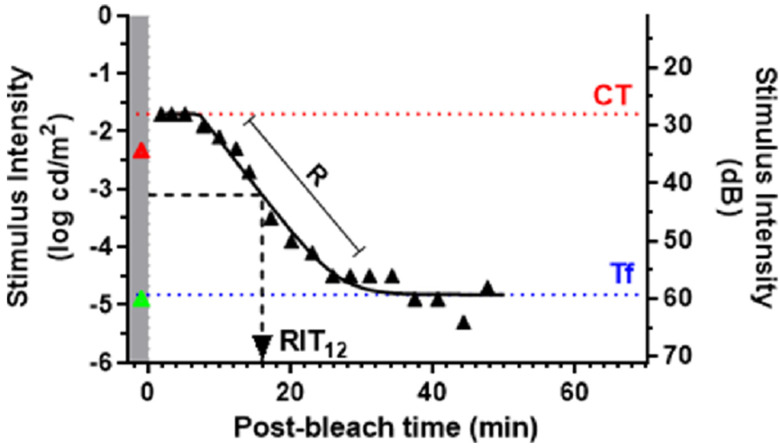
The Dark Adaptation curve reproduced with permission from Uddin et al. [38] This shows a representative dark adaptation curve along with key derived parameters. The gray region with the green and red triangles illustrates the measurement of scotopic thresholds at 505 nm and 625 nm respectively before delivery of bleach. Following exposure to a bleaching light, retinal sensitivity reaches an initial plateau mediated by cones (CT). Once rods become more sensitive to cones (Cone-rod break), retinal sensitivity continues to improve at a fixed rate (R dec/min) before reaching final asymptotic rod threshold (Tf). Rod intercept time (RIT), the time taken to detect a stimulus of −3.1 log phot cd/m^2^ has been widely used to assess DA in AMD.

**Figure 3 jcm-11-01358-f003:**
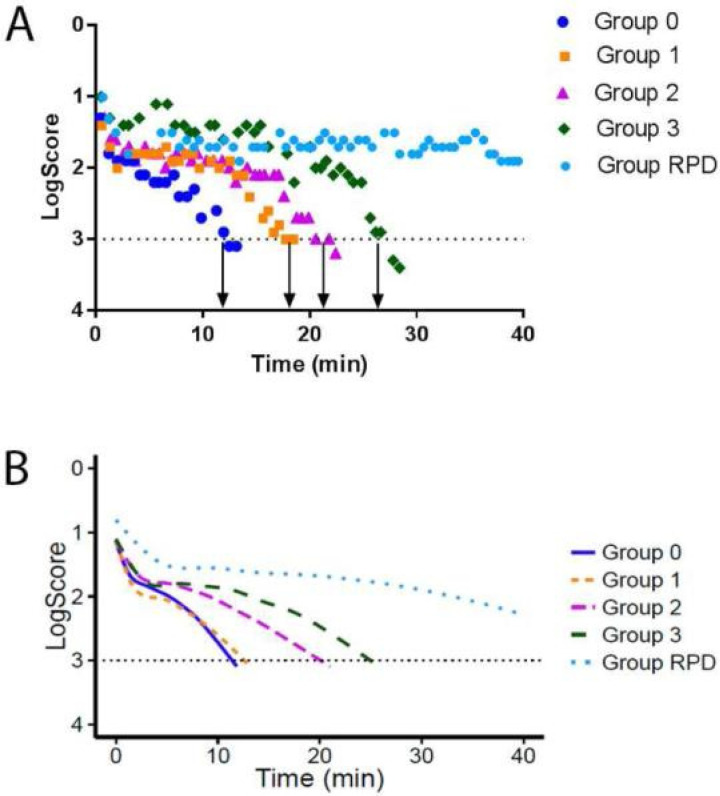
Impaired DA in different AMD stages. Reproduced with permission from Flamendorf et al. [98], an example for impaired DA in different AMD stages. (**A**). Representative dark adaptation raw data for individual participants with no large drusen (group 0), large drusen in the study eye only (group 1), large drusen in both eyes (group 2), advanced disease in the nonstudy eye (group 3), and reticular pseudodrusen (RPD). The rod intercept time is the time required for the patient’s visual sensitivity to recover to a stimulus intensity 3 log units dimmer than the initial threshold (arrows). (**B**). Graph showing averaged raw data for each group. The group curves were derived by averaging the fitted values over a grid of points (2-s intervals) from time 0 to 40 min based on a 3-component piecewise linear fit to the raw data (excluding fixation errors).

**Figure 4 jcm-11-01358-f004:**
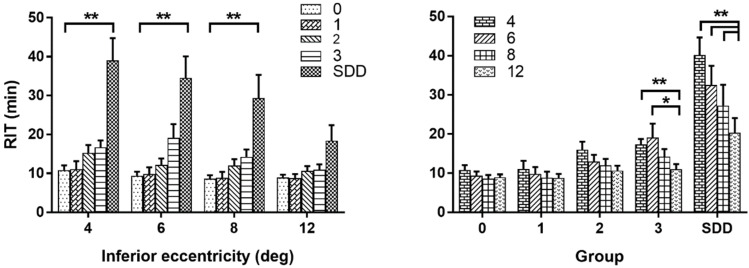
Dark adaptation in different AMD disease stages and on various retinal locations in the inferior retina in the vertical meridian. Reproduced with permission from Flynn et al. [54]. The left figure shows mean rod intercept time (RIT) for different eccentricities, showing that alterations are more pronounced in the 4°, 6° and 8° compared to the more eccentric 12° locations. (Post hoc comparisons to group 0 by eccentricity: ** *p* < 0.0001) The right figure compares different disease stages at certain retinal locations exhibiting markedly increased RIT values in presence of SDD. (Post hoc comparisons of RIT at 4°, 6° and 8° relative to 12° by group; * *p* < 0.0015, ** *p* < 0.0001).

**Table 1 jcm-11-01358-t001:** Instruments for DA measurement.

Instrument/Model	Type/Primary Use	Instrument Output
Goldman-Weekers	Perimetry (modified to measure DA)	DA curve–DA parameters derived after offline data processing
Modified Humphry Filed Analyzer *	Perimetry (modified to measure DA)	DA curve–DA parameters derived after offline data processing
Metrovision MonCvOne *	Perimetry (built-in DA measurement functionality)	DA curve–DA parameters, including RIT, final threshold etc. Raw data accessible
Medmont Dark Adapted Chromatic Perimeter *	Perimetry (DA measurement requires user supplied system for delivery of background light for rhodopsin bleaching	DA curve and parameters derived after offline data processing. Raw data accessible
Roland Consult DARK-adaptometer *	Perimetry (built-in DA measurement functionality)	DA curve–DA parameters derived after offline data processing
Maculogix AdaptDx Tabletop *	Dedicated DA measurement	Rod-intercept time with DA curve up to a criterion threshold. Fixation error percentage, Raw data accessible
Maculogix Adapt Dx Pro Wearble, with inbult tracking, artificial intelligence guided	Dedicated DA measurement for screening	Rod-intercept time and percentage of fixation errors, machine data not accessible
Heru re:Vive 2.0	Dedicated DA measurement for screening	

* Instrument for clinic and research protocol with adjustable test duration and spot location.

## Data Availability

Not applicable.

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
