# Peer review of "Dark Adaptation and Its Role in Age-Related Macular Degeneration"

_jcm, 2022, doi:10.3390/jcm11051358_

Round 1

Reviewer 1 Report

Overall, this is an excellent review of dark adaptation, both the biochemical basis as well as its potential role as a biomarker for AMD.  The manuscript is very comprehensive, presenting both a comprehensive review of some of papers that recognition of abnormal of dark adaptation in the AMD process as well as an overview of some of the more "modern" use of specific DA parameters that can be correlated to the stage of AMD.  If I would add anything, it would be some brief statement about the potential that DA analysis can be used by practitioners in their day-to-day practice (ie what are the obstacles to having it be used more commonly, maybe side by side with oct to stage AMD and assess its progression - is it still acquisition time, billing, etc?).  Otherwise, I thought this was very well written. 

Author Response

  • Reviewer 1
  • Comments and Suggestions for Authors
  • Overall, this is an excellent review of dark adaptation, both the biochemical basis as well as its potential role as a biomarker for AMD.  The manuscript is very comprehensive, presenting both a comprehensive review of some of papers that recognition of abnormal of dark adaptation in the AMD process as well as an overview of some of the more "modern" use of specific DA parameters that can be correlated to the stage of AMD.  If I would add anything, it would be some brief statement about the potential that DA analysis can be used by practitioners in their day-to-day practice (ie what are the obstacles to having it be used more commonly, maybe side by side with oct to stage AMD and assess its progression - is it still acquisition time, billing, etc?). Otherwise, I thought this was very well written.

We appreciate this suggestion and have added Section 2.4 Use of dark adaptation as a screening tool for AMD, page 19, Line 827-848

“2.4 Use of dark adaptation as a screening tool for AMD

Two studies have investigated the clinical use of the AdaptDx for diagnosis and prognosis of AMD.  In a study utilizing a short (<6.5 minute) DA test in participants with intermediate AMD and no AMD, a binary interpretation of the DA test was found to have a 90% diagnostic sensitivity and specificity for diagnosing AMD compared to gold standard color fundus grading [31]. A subsequent longitudinal study suggested that aged eyes with ab-normal DA are more likely to develop AMD (as graded using color photographs) 3 years later [34].   Whether functional DA changes precede or predict structural AMD changes is under current investigation using additional imaging modalities such as OCT. [26,58,59,65-68].

Several DA instruments now have FDA approval or clearance for use in clinical DA testing.  DA manufacturers have entered the commercial market and promote their instruments to be used outside of clinical trials. (https://www.maculogix.com/amd/dark-adaptation-testing/) (https://www.seeheru.com/technology/) Recent developments include the size and interface of the devices, e.g., as augmented reality glasses (https://www.maculogix.com/adaptdx/Maculogix AdaptDx Pro) and multiple additional functions (https://www.seeheru.com/technology/Heru re:Vive 2.0).  There is also a Current procedural terminology (CPT) code for dark adaptation testing (CPT92284) which is an active code given by the center for Medicare & Medicaid services (CMS) Reimbursement making reimbursement for DA testing is possible under certain indications in the United States including AMD, night blindness, hereditary retinal dystrophy. The reimbursement for AMD is payor specific by insurance providers, however most payors will pay one occurrence per visit without the use of laterality modifiers.”

Deeba Husain

Reviewer 2 Report

Dear authors,

the manuscript well describes the role of dark adaptation in many retinal diseases and its significance.

The literature review and the clinical data you reported are complete and detailed.

The clinical application is reported ad well as the instruments to measure its modifications. I think some more considerations could be added to the real life applications in in-patients and out-patients settings or otherwise if it could find application only in clinical trials tests.

The figures and tables and complete and detailed, from the description of the biological mechanisms to the physiological aspects.

Best regards

Author Response

Reviewer 2

Dear authors,

the manuscript well describes the role of dark adaptation in many retinal diseases and its significance.

The literature review and the clinical data you reported are complete and detailed.

The clinical application is reported ad well as the instruments to measure its modifications. I think some more considerations could be added to the real life applications in in-patients and out-patients settings or otherwise if it could find application only in clinical trials tests.

 We thank the reviewer for this suggestion and have added Section 2.4 Use of dark adaptation as a screening tool for AMD, page 19, line 827-848.

“2.4 Use of dark adaptation as a screening tool for AMD

Two studies have investigated the clinical use of the AdaptDx for diagnosis and prognosis of AMD.  In a study utilizing a short (<6.5 minute) DA test in participants with intermediate AMD and no AMD, a binary interpretation of the DA test was found to have a 90% diagnostic sensitivity and specificity for diagnosing AMD compared to gold standard color fundus grading [31]. A subsequent longitudinal study suggested that aged eyes with ab-normal DA are more likely to develop AMD (as graded using color photographs) 3 years later [34].   Whether functional DA changes precede or predict structural AMD changes is under current investigation using additional imaging modalities such as OCT. [26,58,59,65-68].

Several DA instruments now have FDA approval or clearance for use in clinical DA testing.  DA manufacturers have entered the commercial market and promote their instruments to be used outside of clinical trials. (https://www.maculogix.com/amd/dark-adaptation-testing/) (https://www.seeheru.com/technology/) Recent developments include the size and interface of the devices, e.g., as augmented reality glasses (https://www.maculogix.com/adaptdx/Maculogix AdaptDx Pro) and multiple additional functions (https://www.seeheru.com/technology/Heru re:Vive 2.0).  There is also a Current procedural terminology (CPT) code for dark adaptation testing (CPT92284) which is an active code given by the center for Medicare & Medicaid services (CMS) Reimbursement making reimbursement for DA testing is possible under certain indications in the United States including AMD, night blindness, hereditary retinal dystrophy. The reimbursement for AMD is payor specific by insurance providers, however most payors will pay one occurrence per visit without the use of laterality modifiers.”

The figures and tables and complete and detailed, from the description of the biological mechanisms to the physiological aspects.

Deeba Husain

Reviewer 3 Report

Authors in the manuscript titled “Dark adaptation and its role in age-related macular degeneration” reviewed the studies on dark adaptation measurement and related point with the pathogenesis of age-related macular degeneration (AMD). The manuscript is well written and systematic review on dark adaptation, its measurements, and its application to AMD.

Minor critics due to multiple typo errors

- Lack or improper insertion of period at some sentences: for example, lines 129, 148.

- Application of capital letter: for example, line 782.

- Multiple wrong spacing words throughout the whole manuscript.  

Author Response

Reviewer 3

Comments and Suggestions for Authors

Authors in the manuscript titled “Dark adaptation and its role in age-related macular degeneration” reviewed the studies on dark adaptation measurement and related point with the pathogenesis of age-related macular degeneration (AMD). The manuscript is well written and systematic review on dark adaptation, its measurements, and its application to AMD.

We thank the reviewer for their review of our article.

Minor critics due to multiple typo errors

- Lack or improper insertion of period at some sentences: for example, lines 129, 148.

 We agree with the comment and have inserted periods at line 129 and removed extra period at line 148 period

- Application of capital letter: for example, line 782.

 We thank the reviewer for detailed review and appreciate the suggestion. The sentence is changed adding capital letter D to ‘Despite’ to, - “However, it is also important to consider and investigate how these measures, and in particular, dysfunction, impact the participant’s activities of daily living. Despite a good visual acuity, these impairments can be not only uncomfortable but, in some situations - dangerous (e.g., tunnels when driving).”

- Multiple wrong spacing words throughout the whole manuscript.

We agree with the wrong spacing between multiple words and greatly appreciate the suggestion. The spacing is adjusted on following lines, 22, 24, 30, 54, 60, 87, 104, 112, 139, 157, 176, 178, 183, 209, 211, 254, 256, 279, 307, 319, 395,  448, 450, 457, 459, 461, 481, 484, 488, 508, 512, 520, 521, 524, 525, 533, 536, 562, 564, 611, 613, 615, 617, 619, 620, 622, 624, 635, 654, 658, 664, 700, 720, 739, 781, 790,

Apart from the above, following additional edits were made at the line numbers, -

Line 4 symbol added,

Line 28 commas deleted,

Line 117 period insertion corrected,

Line 315 article ‘the’ added,

Line 377-378 grammar corrected,

Line 390 year changed to years.,

Line 392-395 Edits added,

Line 406 period inserted,

Line 517 grammar edit,

Line 532 article added,

Line 533 grammar edit,

Lines 866 and 867 information edited.

Deeba Husain